# Metagenomics Next Generation Sequencing (mNGS): An Exciting Tool for Early and Accurate Diagnostic of Fungal Pathogens in Plants

**DOI:** 10.3390/jof8111195

**Published:** 2022-11-13

**Authors:** Fatma Şeyma Gökdemir, Özlem Darcansoy İşeri, Abhishek Sharma, Premila N. Achar, Füsun Eyidoğan

**Affiliations:** 1Plant, Soil and Microbial Sciences, Michigan State University, East Lansing, MI 48824, USA; 2Department of Molecular Biology and Genetics, Faculty of Science and Letters, Başkent University, Ankara 06790, Turkey; 3Institute of Food, Agriculture and Livestock Development, Başkent University, Ankara 06790, Turkey; 4Amity Food and Agriculture Foundation, Amity University, Noida 201313, Uttar Pradesh, India; 5Department of Molecular and Cellular Biology, Kennesaw State University, Kennesaw, GA 30144, USA

**Keywords:** mNGS, metabarcoding, ITS, bioinformatic analysis, fungal pathogens

## Abstract

Crop output is directly impacted by infections, with fungi as the major plant pathogens, making accurate diagnosis of these threats crucial. Developing technology and multidisciplinary approaches are turning to genomic analyses in addition to traditional culture methods in diagnostics of fungal plant pathogens. The metagenomic next-generation sequencing (mNGS) method is preferred for genotyping identification of organisms, identification at the species level, illumination of metabolic pathways, and determination of microbiota. Moreover, the data obtained so far show that this new approach is promising as an emerging new trend in fungal disease detection. Another approach covered by mNGS technologies, known as metabarcoding, enables use of specific markers specific to a genetic region and allows for genotypic identification by facilitating the sequencing of certain regions. Although the core concept of mNGS remains constant across applications, the specific sequencing methods and bioinformatics tools used to analyze the data differ. In this review, we focus on how mNGS technology, including metabarcoding, is applied for detecting fungal pathogens and its promising developments for the future.

## 1. Introduction

Farmers worldwide have struggled with crop losses caused by pathogens, including bacteria, viruses, and fungi. The main biotic stress that causes the most economic damage and losses is fungal pathogens. Although the course of the disease and the loss of crops vary according to the host plant, sometimes, up to 100% crop losses are experienced. These losses will pave the way for alleviating food shortages and ecological degradation in the future. Difficulties in culturing and diagnosing organisms are at the forefront of the unavoidable reasons for yield losses. Therefore, it is crucial to have state-of-the-art methods for detecting pathogens and preventing diseases, aiming to reduce crops losses at all stages of crop production (from growth through harvest and postharvest processing) and to ensure agricultural sustainability. Metagenomics is the most direct and unbiased technique to investigate the microbiomes’ functionality, and it is a relatively new addition to the molecular toolkit for pathologists. The term refers to the practice of randomly sequencing the genomic DNA of samples (crop or soil) in an environment, as in the present study [1,2,3]. Subsequently, the development of gene expression techniques that enable the discovery of new genes and metabolic products inspired the “metagenomic” science, which provides all genomic information that can be obtained without culturing under in vitro conditions.

DNA sequencing approaches provide basic information about the diversity of living things of biological importance. Despite their high cost, Sanger sequencing technologies are one of the most preferred methods in sequencing technologies. However, as an alternative to this; many sequencing technologies are widely used, including third or next generation sequencing technologies (NGS) such as Illumina, Ion Torrent, HeliScope, Pacific Biosciences (PacBio), 454/Roche, Sequencing by Oligo Ligation Detection (SOLiD), and Oxford Nanopore. It is preferred, and reduces the high sequencing cost [4]. Next-generation sequencing technologies enable the sequencing of part or all of an organism’s genome. However, mNGS, which includes third-generation technologies, also allows us to learn about living variance and population genetics. Moreover, metagenomic next-generation sequencing (mNGS) can be used to provide information on the diversity of biologically important resources, analyze DNA sequences, uncover details of metabolic pathways, identify homology-based genes, discover industrially important enzymes, and solve important problems such as the detection of viral and fungal pathogens, among others. 

mNGS technologies are now regularly employed to assess the phylogeny and functionality of non-cultivable microbes, though human pathogens take precedence over plant pathogens. Although metagenomic sequencing technologies have just begun to be used in plant sciences, promising results have been obtained for the future and are beginning to gain importance in agronomic sciences. This study explains how mNGS technology is used in fungal pathogens detection in agronomic sciences, and the review is the result of comprehensive investigation into the potential advantages of a method that may one day be extensively employed for the purpose of identifying fungal pathogens. It is essential to highlight that mNGS by itself does not establish pathogenicity. It helps us to identify plant pathogens that cannot be cultivated in the laboratory. In order to establish pathogenicity, it is necessary to identify the nucleic acid of such pathogens in host tissues and to mutate the genes associated with the virulence. Returning the normal gene to the mutant microbe should restore its pathogenicity. The mutated organism should be less capable or incapable of causing disease.

The search of the literature was performed using important databases such as Web of Science, Springer link, and Scopus. Keywords such as mNGS, next-generation sequencing, fungal plant pathogens, and phytopathogens were used for the literature research. The most recent and up-to-date studies were kept as a priority. Although the use of NGS studies dates back a long time, studies involving metagenomic NGS analysis are still in their infancy in the agricultural sciences, particularly, in the detection of fungal diseases. For this reason, the importance of mNGS technologies in terms of methodological approach in agronomic sciences has been emphasized in our study, and has a high potential to be used as a common trend in the future.

## 2. Multiple Real-World Applications for mNGS

mNGS technologies can be optimized for use in many areas today (Figure 1). Even if each usage area seems different, mNGS is a common point due to the similarity of the specific barcodes and the method used (Figure 2). One of the primary purposes of mNGS is detecting all culturable and non-culturable substrates and organisms in the medium or host. For this reason, organisms can be scanned via barcodes specific to the species to be determined. There are 16 S rRNA-based universal barcodes used for bacteria, while barcodes from the ITS region are preferred for fungal pathogens. Evolutionary and ecological studies have a vital role in the development of metagenomic science. The first discovery of proteorhodopsin proteins occurred in environmental DNA. Complete genome data of microbial communities found in environmental samples can be obtained today, with scientists aiming to reveal whole genomes. Environmental genomes obtained in this way allow us to decipher the details of organisms’ metabolic pathways and create a gene inventory. Environmental DNA or mixed DNA samples help us to understand the genetic microheterogeneity of bell groups [5].

According to the review of the literature, metabarcoding or metagenomic sciences have been widely used in health sciences up to now [6]. The first use of metagenomics in health sciences dates back to 2008 [7]. After the organ transplant of three different patients, the accompanying analysis showed Arenavirus in recipients using the mNGS method. Following this report, mNGS became a routinely accepted method for detecting infectious diseases to date [8]. By analyzing body fluids [9], detecting pulmonary infection in lung tissues [10,11] and microbial organisms underlying chronic meningitis, determining organisms causing tuberculous meningitis in cerebrospinal fluid [12], and even identifying pathogens responsible for uncultured prosthetic joint infection [13] have become trends of choice. Most of the studies use viral, bacterial, and fungal kits. In addition to the detection of different infections, scientists aimed to map human-associated microbial communities, such as the gut, mouth, skin, and vagina, as part of the Human Microbiome project [14,15]. mNGS technology is also used in forensic sciences, particularly in the resolution of forensic cases such as geographic locations and surface analysis [16,17], identification [18,19], biological sex determination [20,21], trace evidence [22], manner and cause of death determination [23,24], and postmortem microbiota determination are becoming more and more common [25,26].

mNGS technologies in the agricultural and industrial fields have led to important discoveries. New generation sequencing studies, primarily available in plant roots, are increasingly preferred, as they enable the discovery of important secondary metabolites, enzymes, and metabolites [27]. With the influence of industrial applications of the metagenomic approach, the discovery of stress-sensitive bioactive compounds reveals the genetic information of organisms living in extreme conditions. This discovery is used for efficient crop production and elucidation of plant stress mechanisms.

Agronomically, the scope of mNGS technologies is expanding day by day. In a previous report, microbial diversity data are essential for sustainable black pepper production [28]. The organisms that make up the plant microbiota provide the necessary nutrients for the growth and development of the plant. Therefore, mNGS technologies are vital for the sustainability of agriculture. Moreover, using metagenomic data to detect and control biotic stress factors affecting crop yield offers optimistic promises for the future. For example, the metagenomic method with 16 S rRNA barcodes was applied to samples obtained from black pepper roots grown in Vietnam [28], with promising outcomes.

## 3. mNGS Methodology for Detecting Fungal Pathogens in Plants

### 3.1. Wet Lab Applications

Obtaining a suitable sample is important for mNGS technology to be applicable. In detecting latent pathogens, it is necessary to use plants that are still alive, but highly infected. When taking the sample, the plant should be selected where the infection symptom is most evident. The conditions of infection for latent pathogens may differ according to the experimental design. For instance, if the study aims to determine an infection in plants in an uncontrolled area, metagenomic sampling should be differentiated based on the symptoms expressed by the microorganism. Fungal stress is the main biotic factor that causes a considerable decrease in yield. Viruses and bacterial infections are also common in plants under natural conditions [29]. Identifying a fungal pathogen with bacterial barcodes is pointless. The distinction between which abiotic stress causes infection in the plant should be made with observational techniques [30]. Collected samples should be kept in a cold environment (4 °C in the refrigerator), transported to the laboratory environment, and stabilized. Storing experimental samples under standard ambient settings for an extended period poses a risk of DNA contamination from other organisms. This may compromise the sensitivity of metagenomic analysis and lead to misinterpretation of the data [31].

Nucleic acid extraction is the initial step of mNGS analysis. Extraction can be performed using either commercial kits or standard manual procedures, but the former is recommended in order to rule out the possibility of environmental contamination. Extraction experiments should be performed in an aseptic environment. Since the extracted nucleic acids will comprise DNA from multiple species, they are referred to as mix-DNA, and if they are collected from environmental samples, they are known as environmental DNA or eDNA [32]. Sometimes, traditional culturing methods can be used to confirm latent infection. This will ensure that the dominant pathogen in the plant is reproduced in vitro, and it will be possible to determine whether the plant is indeed an organism-borne infection. It can be considered as a control mechanism for metagenome sequencing. However, this is optional. DNA isolation can also be performed directly from the infected leaf using suitable kits.

### 3.2. Preparation of Library

The purpose of not preparing a library for mNGS is to make the resulting nucleic acid mixture compatible with sequence analysis. While preserving the diversity of DNA sequences in microbiota analysis, it is necessary to protect or enrich the sequences in pathogen studies. Therefore, the library preparation is a complex process. In some metagenomic analyses, the entire nucleic acid obtained can be sequenced, or strategic barcodes of a particular microorganism population can be used. This is because even the most efficient DNA sequencing technologies can sequence only a small fraction of DNA and RNA [33]. Therefore, the prepared library should be representative of the original sample.

In investigations designed to detect pathogen or microbiota on the plant, it is anticipated that most nucleic acid extracted will be from the plant. However, using the necessary purification kit, the DNA of the pathogen or microbiota can be separated from the plant’s DNA. mNGS libraries can be constructed using minimal amounts of extracted microbial nucleic acid. Microbial enrichment techniques can be used for both DNA and RNA. For the determination of the pathogen, a comprehensive DNA library is created [34]. Pathogenic fungal, bacterial, and viral fragments can be amplified by Polymerase Chain Reaction (PCR) amplification to increase the nucleic acid content of existing pathogen.

When nucleic acid samples are ready for sequencing, sample barcodes and sequencing adapters are added. Barcoding technology involves using short strings of specific markers (barcodes) added to the end of the sample booklet [33]. This allows multiple samples to be used together for sequencing and to generate sample ID for each sequence read determined by bioinformatic analysis. Library preparation kits, such as the high-tech Nextera XT (Illumina, San Diego), are sensitive enough to work with one ng of DNA.

Preparing a library is an important step in identifying metagenome reads for a particular gene region. However, there is no standardized process for preparing a metagenomic library. The library preparation process in existing mNGS studies is carried out with precision kits (Illumina, NEB, etc.) developed by various companies [35].

### 3.3. Sequencing

Various high-throughput platforms are used for the sequencing of mNGS samples. The most used methods in metagenomic studies are Illumina sequencing, Nanopore sequencing, and Roche/454 pyrosequencing. Ilumina sequencing can provide more sensitive and unique results compared to other techniques, with a read depth of 1 to 5 million at 75 to 100 base pair alignments. Specific 16S rRNA barcodes are used to detect bacterial infections, while barcodes used for the ITS-23S rRNA region are used to detect viral organisms [36]. Some studies may require the use of both barcodes in conjunction. In studies where the plant species is unknown, barcodes explicitly defined for the plant can be included in the study by using a method called metabarcoding. The most preferred universal plant barcodes are rbcL, trnL-trnF, rpc36-8, trnT2-rps4, and two mitochondrial genes, nad7 and atpA [37]. 16S rRNA for detection of bacterial organisms, barcodes of ITS, and 18 S rRNA genes for fungi and archaea are preferred.

### 3.4. Bioinformatics Data Analysis

After the metagenomic next-generation sequencing process, a series of bioinformatic analyses is required in order to analyze the data. The hundreds of short reads obtained in the sequencing must first be filtered. The aim is to extract poor-quality sequences and host genome data. To extract short sequence reads, including the plant genome, a comparison with a reference genome is used to extract matched reads [38]. After filtering, the remaining sequences are compared with reference microbial sequence databases. NCBI is the most preferred database, since it is possible to reach genomic data of many organisms to be detected. Large sequence reads are combined de novo in clusters each called a contig, which is derived from the word “contiguous”. A contig, in genomic sequencing, is defined as a set of DNA sequences that overlap, and it provides a contiguous representation of a genomic region enabling links to physical maps. The aim is to assign as many groups as possible to every possible taxonomic group (species, genus, phylum). Reads that do not match any sequence are combined de novo with unique algorithms developed for metagenomics (Table 1). De novo joins can be done with Meta Velvet and Meta-IDBA software [39,40].

## 4. Successful Applications of mNGS in Fungal Plant Pathogen Detection

mNGS technology holds promise for pathogen detection in plants. It is used for definitive disease diagnosis, since it provides sequencing of all nucleic acids in samples taken from infected tissue, regardless of traditional culture methods. Since the barcodes specific to the disease agent are not used, there is no need for pre-sequencing information of the infecting organism. However, it is recommended to use a database with genome information of fungal pathogens while analyzing the results. 

Yang et al. used metagenomic analysis to discover and identify *Calonectria pseudonaviculata*, the fungus that causes boxwood blight in plants [50]. According to data obtained using different DNA isolation protocols and different bioinformatics algorithms, more than 9% of the reads performed in highly-infected plant tissue were identified as *C. pseudonaviculata* [50]. This study shows how metagenomics can be applied to plant pathogens, and this tool is promising for future studies on fungal pathogens. More so, fungal infection is the leading biotic stress factor affecting yield and quality of products in agricultural areas. Unfortunately, approaches to detect plant–fungus interactions at the molecular level are progressing at a slow pace. As we indicated, plant pathogenic fungi are less defined than bacterial and viral infections in databases with genomic data.

One of the pioneering studies in fungal pathogen detection studies identified the microbiome of plants infected by *Zymoseptoria tritici* in wheat plants [51]. The result, obtained in this study using 450 leaf samples, shows significant differences in the microbiota of healthy tissue and infected tissue. However, the microbiomes of infected leaves collected from different cultivars show very high similarities. The collected data can help prevent infection by *Zymosepttoria tritici* and improve wheat health. 

Although sequencing analyses using mNGS technology have become widespread, many approaches using metagenomic techniques have identified the causal agents to be fungi. Some of these results are shown in Table 2.

## 5. Conclusions

The use of mNGS technology to identify fungal pathogens and its relationship with plants is promising for the future. Uncovering the plant–microbiota interaction will, in turn, enable the discovery of new genomic data and new industrially important biological materials. Moreover, its dissemination in agronomic sciences will enable the development of methods to combat biotic stress in food-related problems. Detection and identification of infectious agents in the plant’s phyllosphere region aid in the proper management and control of pathogens, hence boosting agricultural and crop yields.

Through mNGS, economically significant pathogens that cannot be cultivated using standard approaches may be detected and researched. Additionally, related DNA samples (stress-tolerant genes) of plants that may be expressed due the presence of plant pathogens can be studied (interaction study). In the not-too-distant future, the mNGS technique, which is used in addition to standard identification and characterization methods, will become a significant phenomenon and a common practice in agrobiotechnology. Although the metagenomic technology applied today has a high cost, there is also potential to reduce the cost, thanks to the increasing demand and development of technology. As the cost decreases, mNGS technology will become more widespread.

## Figures and Tables

**Figure 1 jof-08-01195-f001:**
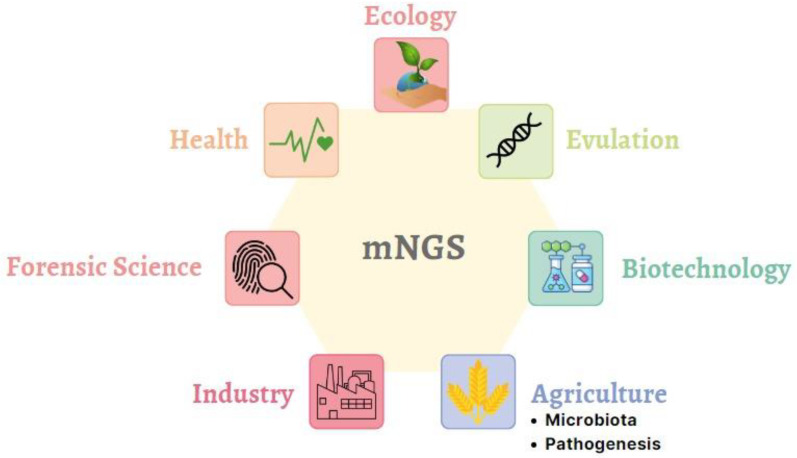
Applications of mNGS technology in different fields.

**Figure 2 jof-08-01195-f002:**
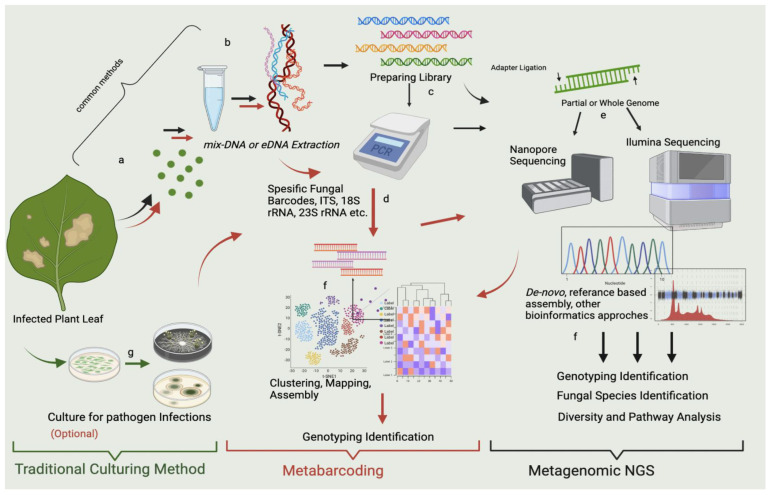
mNGS and metabarcoding workflow chart for the sample obtained from the infected leaf. The workflow highlighted in red shows metabarcoding pathways, which use specific metabarcodes for fungal detection, and the “black” arrow shows the mNGS pathways. In the workflows, the PCR stage is optional, and after sequencing and bioinformatic analysis, metabarcoding shows genotyping identification. mNGS indicates fungal species identification, microbial diversity, pathway detection, and genotyping identification. Both techniques seem to include the same steps; however, the algorithms (the bioinformatics analysis) differ. In metabarcoding, certain parts of the genome are sequenced using target-specific barcodes, and in mNGS, either a partial or a whole genome is sequenced by reference-based comparison with the prepared library. Both approaches provide a fundamental approach and solution for metagenomics. The workflow highlighted in green represents the traditional culturing method at the researcher’s discretion. It may allow culturing of some of the possible microorganisms prior to mNGS and metabarcoding. However, this gives an assignment far below sufficient for mNGS and metabarcoding. The stages represented in the figure can be summarized as follows: (a) sampling of infected parts of the plant (leaf discs are preferred); (b) DNA extraction from leaf; (c) library preparation; (d) PCR amplification of gene regions of microbial pathogens with specific gene barcodes; (e) sequencing with Illumina, Nanopore, etc.; (f) bioinformatic analysis of mNGS containing de novo approaches and referenced based assembly, bioinformatic analysis for metabarcoding assembly, clustering, and prediction; and (g) control culture of infected leaves.

**Table 1 jof-08-01195-t001:** Algorithm tools (Bioinformatics Analysis) for the post-mNGS process.

Purpose	Algorithm Tools	References
OTU clustering	MOTHUR, SUMACLUST, SWARM, METACLUSTER, UCLUST, CD-HIT-OUT, TBC	[41,42]
Phylogenetic classifications	Phymm, BLAST, CARMA	[43]
Denoising	Pyronoise, Denoiser, DADA, Acacia	[44,45]
Chimera detection	UCHİME, ChimeraSlayer, Persus, DECIPHER	[45,46]
ITS database for fungal detection	UNITE	[47]
All in one	MOTHUR, QIIME, MEGAN	[45,48,49]

**Table 2 jof-08-01195-t002:** The successful applications of metagenomic techniques to diagnose fungal pathogens.

Plant	Aim of Study	Metagenomics Techniques	References
Grape	Determination of fungi and oomycetes in different phyllosphere samples	Metabarcoding	[52]
Grape	Determination of soil and leaf-associated fungal microbiota	mNGS-Ilumina	[53]
Wheat	Detection of fungal microorganisms in the wheat phyllosphere	Microbiome Metabarcoding using ITS barcodes	[54]
Grape	Identification of fungal diseases on the vine trunk	mNGS-Ilumina	[55]
Maize	Determination of fungal microbiota after harvest	Metabarcoding	[56]
Wheat	Determination of fungal communities in wheat residues	Metabarcoding	[57]
Grapevine	Determination of fungal disease agents associated with grapevine	Metabarcoding	[58]
Banana	Investigation of the effect of variable soil microbiota on fusarium disease	Metabarcoding	[59]
Wheat, maize	To determine *Fusarium* species in various plants	PaCBio SMRT Sequencing	[60]
Strawberry	Determination of microbial communities in strawberry growing soils with different yields	Amplicon Based Metagenomic	[61]

## Data Availability

All the data are present in the manuscript.

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
