# Peer review of "Metagenomics Next Generation Sequencing (mNGS): An Exciting Tool for Early and Accurate Diagnostic of Fungal Pathogens in Plants"

_jof, 2022, doi:10.3390/jof8111195_

Round 1

Reviewer 1 Report (Previous Reviewer 3)

The work presented to me for review, entitled "Metagenomics Next Generation Sequencing (mNGS): An Exciting Tool for Early and Accurate Diagnostic of Fungal Pathogens in Plants" has been greatly improved. The authors complied with all my comments, which contributed to a significant improvement in the quality of the article. The number of citations has also increased significantly, although it still appears to be small when considering a review. My concerns are also caused by the volume of the entire article. Excluding figures and tables, it is only 5 typescript pages. Unfortunately, I still have the feeling that the presented work is nothing more than a broader extension of the "methodology" section of a standard research article. My Comment "However, I have a very serious problem with determining whether this article brings anything new. Provides an overview of the basic information related to the potential use of mNGS and the description of the methodology. Similar information in a slightly smaller form can be found in many original research where mNGS was used for analysis." is still remains valid and therefore I cannot rate the article well under the terms of “Is the work a significant contribution to the field?”. However, perhaps this form of work will contribute to the dissemination of mNGS methods in agriculture and will serve as a ready methodology for the application of this technology.

Author Response

We are grateful for the time and work that substantially improved our manuscript. Since there has been no significant study on mNGS in plant pathology, we could not extend the scope of this paper beyond a certain point. However, as noted by a distinguished reviewer, the purpose of this paper is to promote the use of mNGS techniques in agriculture. We believe it will be a ready-made approach for implementing this technology in the agricultural sector.

Reviewer 2 Report (Previous Reviewer 2)

Dear Authors of the manuscript "Metagenomics Next Generation Sequencing (mNGS): An Exciting Tool 2 for Early and Accurate Diagnostic of Fungal Pathogens in Plants", the adjustments and responses provided to the reviewers' comments were appropriate.

The re-submitted manuscript has been improved and is suitable for publication.

Kind regards,

Reviewer 2.

Author Response

Thank you so much! We appreciate your comments that helped us in improving the manuscript. 

This manuscript is a resubmission of an earlier submission. The following is a list of the peer review reports and author responses from that submission.

Round 1

Reviewer 1 Report

The manuscript entitled "Metagenomics Next Generation Sequencing (mNGS): An Excit-ing Tool for Early and Accurate Diagnosing Plant Fungal Pathogens " has been reviewed.

1.      The review seems a method introduction. The authors should give more information of the advantage examples after applying mNGS to detect fungal pathogen or microbiota.

2.      There are many ‘pathogenicity’ in the manuscript. Is there any mistakes? The author should focus on the ‘fungal pathogen’.

3.      There are some places need to be revised. Please check the attachment.

Reviewer 2 Report

Dear authors of the manuscript "Metagenomics Next Generation Sequencing (mNGS): An Exciting Tool for Early and Accurate Diagnosing Plant Fungal Pathogens”, after evaluating your review, I recommend the following adjustments to improve it:

- Page 1, line 17: Replace “Crop output is directly impacted by plant infections (with fungi as the major pathogen),” to “Crop output is directly impacted by plant infections, with fungi as the major pathogen,”.

- Page 1, line 40: Add “pathogens” after “detecting”.

- Page 1, line 40: Replace “if we reduce the loss” to “aiming to reduce losses”.

- Page 1, lines 42-44: Please, review whether the causes and consequences presented in the sentence are adequate.

- Page 2, line 72: Add “and substrates” after “hosts”.

- Page 2, line 75: Perhaps it is more appropriate to write that ITS region is preferred for fungi, rather than fungal pathogenicity, which may imply that the ITS region contains specific factors related to pathogenicity.

- Page 3, line 113 and where else in the text: would “latent” be a more appropriate word for “vegetative”?

Table 1. Provide a footnote with the meanings of abbreviations, as tables should be self-explanatory.

- Page 6, line 220: Are they really rare? In relation to which organisms? clarify.

- Page 8, line 253: Complement “are essential in” with “for the proper management and control of pathogens and as a consequence…”.

- Page 1, lines 257-259: Please review and improve the writing of the two sentences presented on the indicated lines.

Kind regards,

Reviewer.

Reviewer 3 Report

The article presented to me for review, entitled "Metagenomics Next Generation Sequencing (mNGS): An Exciting Tool for Early and Accurate Diagnosing Plant Fungal Pathogens", raises an issue that is very important from the point of view of people dealing with plant pathology, related to the rapid diagnosis of diseases that cause damage to crops. Certainly, the use of metagenomics next generation sequencing is a great facilitation in everyday diagnostic work. However, I have a very serious problem with determining whether this article brings anything new. Provides an overview of the basic information related to the potential use of mNGS and the description of the methodology. Similar information in a slightly smaller form can be found in many original research where mNGS was used for analysis. The entire work consists of only 47 literature items, which is a small number when it comes to review work. What is more, the authors did not indicate in the text how the search for relevant works was carried out in some way, which in my opinion should be implemented in the methodology section. For a literature review it is a minimum requirement that you explain how you conducted the literature search. Typically, this is done by doing a comprehensive search using Web of Science or Scopus databases using a combination of relevant search terms and keywords. In addition, one consults the relevant literature by backwards and forward searches for relevant papers that were not revealed by the literature databases. Please add a methods section that explains all this.

Another serious complaint is the way the necessary information is presented. Reading the work, one gets the impression that most of the threads are treated very briefly, and some of them are only mentioned and, in my opinion, should be discussed in more detail. For example: What if the presence of an organism, such as Fusarium, is confirmed, which in various situations may be a pathogen or a saprotroph. Will the mNGS help to distinguish whether an expert analysis is still needed? It seems to me that information on this subject (pathogenicity of the organism in relation to the host plant) should be more emphasized in the text. It is true that at the end of the work, the authors raise the topic of pathogenicity, but it is still not clear whether, without meeting Koch's postulates, we can say on the basis of mNGS that a given organism is pathogenic to the plant under study. Another example is the lack of detailed information in Chapter 3. NGS Methodology for Detecting Fungal Pathogens in Plants. The authors, writing about collecting environmental samples, state: "Collected samples should be kept in a cold environment, transported to the laboratory environment, and stabilized." In my opinion, such phrases should not be used in scientific works. Similar information should be presented more precisely. Ultimately, when reading such a work, it is not known whether the mentioned "cold environment" is, for example, 5 or 10 ° C.

To sum up, the thesis raises very important issues related to the use of mNGS techniques, but it should be re-edited, written in more detail, taking into account the additional chapter on methods of searching for works for the presented review.